# Validation of an Abbreviated Scale of the CENVI Questionnaire to Evaluate the Perception of School Violence and Coexistence Management of Chilean Students: Differences between Pandemic and Post-Pandemic

**DOI:** 10.3390/bs13080686

**Published:** 2023-08-17

**Authors:** Flavio Muñoz-Troncoso, Isabel Cuadrado-Gordillo, Enrique Riquelme-Mella, Gerardo Muñoz-Troncoso, Edgardo Miranda-Zapata, Karina Bizama-Colihuinca, Ekaterina Legaz-Vladímirskaya

**Affiliations:** 1Facultad de Educación, Universidad Católica de Temuco, Temuco 4810296, Chile; kbizama@uct.cl; 2Facultad de Educación y Psicología, Departamento de Psicología y Antropología, Universidad de Extremadura, 06071 Badajoz, Spain; cuadrado@unex.es; 3Facultad de Ciencias Sociales y de Artes, Universidad Mayor, Temuco 4801043, Chile; 4Facultad de Filosofía y Humanidades, Universidad Austral de Chile, Valdivia 5110566, Chile; gerardo.munoz01@uach.cl (G.M.-T.); ekaterina.legaz@uach.cl (E.L.-V.); 5Facultad de Psicología, Universidad Autónoma de Chile, Temuco 4810101, Chile; edgardo.miranda.z@uautonoma.cl; 6Laboratorio de Aprendizaje Basado en la Comunidad, Universidad Católica Silva Henríquez, Santiago 8330226, Chile; 7Facultad de Educación, Universidad San Sebastián, Valdivia 5110693, Chile

**Keywords:** school violence, school coexistence, coexistence management, intercultural education, racism, discrimination, Mapuche, colonialism, questionnaire, emotions, measurement invariance, measurement model, pandemic, post-pandemic

## Abstract

The objective of the study was to specify an abbreviated model of the school coexistence questionnaire for non-violence (CENVI) for students from 5th to 8th grade (9 to 14 years old), in order to determine the perception of violence and management of school coexistence, and the differences between Mapuche and non-Mapuche students. A total of 1870 students from schools in the city of Temuco (Chile) responded to the CENVI questionnaire. There were two samples: (1) Pandemic, with online, face-to-face and hybrid classes; and (2) Post-pandemic, with face-to-face classes. Sample 1 consisted of 848 students aged 9 to 15 years (M = 11.90; SD = 1.27). Sample 2 consisted of 1022 students aged 9 to 14 years (M = 11.46; SD = 1.14). The questionnaire was validated using expert inter-judgment and Confirmatory Factor Analysis. A good fit of the proposed model to the data and good internal consistency measured according to the composite reliability were found, and convergent validity was demonstrated. Mapuche students perceived more physical violence and social exclusion. Cut-off points were proposed for the interpretation of the results. In the data, the effect of Coexistence Management on School Violence was null. The discussion approaches the findings from the literature on education in spaces of socio-cultural diversity in a Mapuche context.

## 1. Introduction

Several investigations have shown that school violence increased significantly after the return to face-to-face classes, compared to the figures reported during COVID-19 confinement [1,2,3]. In Chile, the last report of the Superintendence of Education (SUPEREDUC) regarding complaints in the area of school coexistence reached 1586 cases, where 40% refer to the mistreatment of students, 17% to disciplinary measures, 14% to discrimination and 4% to participation in school activities [4].

The confinement forced the school system to take an immediate step from teaching in face-to-face spaces to online, telematic or virtual scenarios [5]. Thus, as stated by Cedillo-Ramirez [6], the physical–social distancing induced by confinement, together with the transition to telematic education modalities and the use of recreational spaces through online games, contributed to the increase in digital violence. Likewise, it favored a decrease in other forms of violence that occur only in spaces in which face-to-face interaction takes place. This is in agreement with the study by Jordán and Lira [7], which showed a decrease in victimization and school violence during confinement. The research showed statistically significant differences in this regard between online and face-to-face classes. Likewise, during the pandemic, as a result of confinement, in Latin America there was an increase in gender-based violence in intra-family spaces, which, with respect to the Economic Commission for Latin America and the Caribbean (ECLAC), showed an increase in the incidence of domestic violence [8]. The immediate and long-term consequence of this was an increase in school violence. This is explained by the exposure of children and young people to domestic violence, as spectators and/or victims, who could later replicate and normalize the phenomenon in schools [8].

Violence is referred to as a historical construct associated with the domination of an individual or group towards another individual or group [9,10,11,12,13]. For Galtung [14], it arises from the interaction of direct, structural and cultural violence, where by institutionalizing structural violence and internalizing cultural violence, direct violence will be cyclical and ritual. Hans [15], who incorporates the concept of microphysical violence, argues that it would be shaped by the relationship between internalization, automation and naturalization, which prevents reflection on violence, makes it invisible and does not allow its eradication. From the perspective of Mella et al. [16], violence is multifactorial in origin and involves the victim, the aggressor and the social system. For its part, the World Health Organization (WHO) attributes to it a multicausal origin explained by family, community, cultural and biological factors [17,18]. All in all, violence is a phenomenon that has evolved in conjunction with social systems, transcending diverse scenarios and permeating spaces such as the school system [10].

The relevance of addressing school violence is supported by several studies, which show the impairment of physical and psychological health and social development in children and young people who are participants in the phenomenon, either as victims, aggressors or witnesses [19,20,21,22]. From public policy in education, violence is addressed by taking into consideration that its negative effects on students have detrimental implications on their mental health and learning outcomes [23,24,25]. Therefore, school violence is considered a complex phenomenon, since it is used as an illegitimate strategy for conflict resolution within the school, bringing with it serious consequences [26,27]. In the perspective of Guajardo et al. [28], who address the concept of ‘reality of school violence’, there are internal levels that affect the phenomenon in a hierarchical manner. They then refer to non-hierarchical external levels such as the school system, educational policies and the context. On the latter, López et al. [29] address the phenomenon of school violence by giving special attention to contexts, given in factors that are part of a larger one. They point out that individual characteristics like age and gender are part of classroom factors such as the ‘classroom climate’, which in turn is part of school factors, such as the ‘school climate’. The latter are embedded in cultural factors, such as individuals’ socioeconomic level and ethno-cultural origin.

Chile is a country that recognizes the existence of indigenous peoples as part of the composition of its population, which predate the Hispanic invasion and the creation of the Chilean State [30]. Currently, in Chile, out of a total population of 17,574,003 people, 2,185,792 are recognized as belonging to an indigenous people, which is equivalent to 12.4% of the national total; of these, 79.8% correspond to the Mapuche people [31]. Throughout their history, the Mapuche have been characterized by their resistance to the processes of Spanish conquest and colonization (1536–1810), and to the military invasion of the Chilean State in the so-called War of Pacification of Araucanía (1861–1885) [30]. Although the military defeat by the Chilean army lead to the dispossession of between 90% and 95% of the ancestral territory and the collapse of the family economy [32], the Mapuche implemented practices of resistance and re-existence centered on processes of cultural control [33]. Thus, today, the Mapuche people have a transversal presence at the national level, although with a greater focus on those territories that are part of the *wallmapu* (ancestral territory), particularly the Metropolitan Regions, Araucanía, Los Lagos, Biobío and Los Ríos [31].

### 1.1. School Spaces with Social and Cultural Diversity

The research of Muñoz-Troncoso [34] revealed the existence of statistically significant differences in the perception that Mapuche and non-Mapuche students had of physical and verbal violence, and that indigenous students perceived higher levels of violence. This finding may be related to the development of socioemotional skills in contexts of social and cultural diversity. In this sense, it has been shown that the Chilean school system does not consider the importance of socioemotional factors and their variation among subjects from different cultures [35]. The authors revealed that schools in Chile do not address the social and emotional processes on which cultural interactions are built in the context of diversity and that are part of people’s behavior. In the study by Riquelme et al. [36], the beliefs on which the Mapuche understanding of how to feel is based were found and described; these have been historically ignored by the Chilean educational system, which has opted to maintain a common affective ideal for students, and responds to the parameters of the hegemonic society. Research findings in this regard [36,37] attribute great value to the contributions of the Mapuche family for the identification of the components that make up their belief system.

From this perspective, it is important to make *kimeltuwün* visible as a Mapuche educational action, which contains the epistemological components upon which the education of children and adolescents is implemented within the family and community [38,39,40]. The *kimeltuwün* is based on participation from an early age in the collective activities of the family, and, later, in those of the community. Thus, they learn about the culture and its values via deep observation [41]. Children must learn to express the value of their identity, embody social and cultural roles, know how to relate to nature in all of its expressions, recognize the history of their territory or *lof*, and defend their living space or *wallme* [42].

According to recent research, *kimeltuwün* is systematized in principles, methodologies, methods, and educational purposes [38,39,43]; as one of its central educational principles, it considers the following elements: (1) *yamüwal ta che*, refers to respect being the central axis of human–spiritual interaction; and (2) *the küme che geal*, which positions the affective dimension, kindness and generosity as ideal social behaviors for cohabiting a space in a community. In agreement with the aforementioned authors, it is feasible to assert that in the Mapuche educational framework, a formation of people that seeks the establishment of social interaction dynamics centered on the respect and valuation of the other is favored [44].

However, these training processes contrast with the experiences of schooling that Mapuche people have lived through, since school education was employed by the State as a tool of domination and repression in order to ensure the basic social inclusion of the Mapuche in Chilean society [45,46,47]. Then, far from seeking the educational development of indigenous students, schools with a high enrollment percentage of Mapuche students inserted a school curriculum reduced to the basic rudiments into Mapuche contexts. That is, a curriculum limited to literacy and mathematics, national history, Spanish monolingualism and national identity, as a basis for the formation of the idea of belonging to the State [30,48]. At the international level, different indigenous peoples have lived an experience of schooling marked by a common factor, the dynamics of structural violence as a basis for their inclusion in national societies from a colonialist structure [45,49,50,51].

In the Chilean context, these dynamics of violence and discrimination against the Mapuche were mainly promoted at the beginning of the 20th century. There are several studies that reveal their covert projection in colonialist and monocultural school policies that permeate even intercultural education programs [52]. In this sense, the exclusion of their system of knowledge and own knowledge, and the continuity and creation of prejudices and stereotypes about indigenous people in general and the Mapuche in particular, stand out [53]. Therefore, analyzing the dynamics of school coexistence in the contexts of socio-cultural diversity associated with the Mapuche population implies the recognition of particularities linked to the history of violence and discrimination exercised by the school and its positioning as an instrument of domination, assimilation and impoverishment.

### 1.2. Measuring Perceptions of School Violence and Coexistence Management

Determining the elements of violence and coexistence management present in the Chilean school system is of great importance. Although, internationally, there are several scales for its measurement, one of the most complete existing in Chile is the CENVI questionnaire (CITA). However, its application in schools is difficult because it is extensive, which makes it necessary to develop measurement scales that are simple both in their application and in interpretation of the results, and that measure both concepts together. In addition to the above, it should be considered that the school system omits cultural variations in the emotional development of students, particularly in spaces of social and cultural diversity in a Mapuche context [17,46,48]. Therefore, it is important that the measurement scales are invariant among culturally different groups.

The findings on school violence so far presented are particularly relevant, since this phenomenon has a negative impact on learning outcomes, are the ultimate goal of public policies in the field of education [30,31,32].

In view of the above, the present study aims to specify an abbreviated model of the CENVI questionnaire for the assessment of School Violence and Coexistence Management in the perspective of Chilean students in spaces of socio-cultural diversity in a Mapuche context.

The questionnaire, in addition to being a measurement model that could be used in future scientific studies, is an instrument that can be used locally by school management teams. This questionnaire is performed because it can be used as a diagnostic instrument in schools with similar characteristics to the sample used in this research. Its use and interpretation by schools are able to be incorporated into school coexistence management plans for the prevention of violence. It is worth mentioning that these plans are part of the actions that must be prepared and submitted each year by schools that receive state subsidies [54,55,56]. The present research supports the following hypotheses:

**H1.** 
*The abbreviated version of the scales has good goodness-of-fit indicators, reliability and validity indicators.*


**H2.** 
*The proposed scale has the same meaning for the groups of the categories: gender, ethnicity, dependency and course.*


**H3.** 
*There are statistically significant differences in the perception of violence between Mapuche and non-Mapuche students.*


**H4.** 
*There are statistically significant differences between pandemic participants (online, face-to-face and hybrid classes) and post-pandemic students (face-to-face classes).*


**H5.** 
*There is no causal relationship between the results of Coexistence Management and School Violence.*


## 2. Materials and Methods

This study was based on a research methodology used in psychology and education that is cross-sectional and quantitative in type, with a comparative descriptive design [57,58,59].

### 2.1. Participants

Sample 1 (Pandemic, assessed in December 2021) was composed of 848 students from the commune of Temuco (Chile), aged 9 to 14 years (M = 11.90; SD = 1.27), who were in 5th to 8th grade and attending 4 schools, 2 subsidized private and 2 municipal. Sample 2 (Postpandemia, assessed in November 2022) was composed of 1022 students from the commune of Temuco (Chile), aged 9 to 14 years (M = 11.46; SD = 1.13), who were in 5th to 8th grade and attending 4 schools, 2 subsidized private and 2 municipal. Table 1 shows in detail the characterization of the participants in both samples.

### 2.2. Instrument

The full version of the school coexistence questionnaire for non-violence (CENVI) was used [34]; this explores students’ perceptions regarding types of school violence and coexistence management (Table 2).

The study by Muñoz-Troncoso et al. [34] showed evidence in favor of different elements of validity and reliability. The proposed model of the instrument had a good fit to the data (X2 = 7304.699; DF = 2618; RMSEA = 0.036; CFI = 0.914; TLI = 0.911). The Factor 1 dimensions had correlation measures from 0.6 to 0.9, and the Factor 2 dimensions had correlations from 0.7 to 0.8. All items had factor loadings greater than 0.5 with statistical significance (*p* < 0.001). The composite reliability for each dimension was ω > 0.89. The average variance extracted (AVE) shows values greater than 0.5. Finally, the reliability indicators were optimal, since the dimensions had a value ω ≥ 0.9. Some details of the above are shown in Table 3.

The instrument was composed of a Likert-type self-response scale, in which the students indicated the frequency with which the situations expressed in the items occurred. The indicators of Factor 1 were written inversely and the items of Factor 2 were written directly. The scores were as follows: Never = 1; Seldom = 2; Frequently = 3; and Always = 4. Therefore, in the dimensions of Factor 1, the higher the score, the higher the perception of violence, and in the dimensions of Factor 2, the higher the score, the higher the perception of favorable actions for good school coexistence. The re-specified model [34] presents 2 s-order factors defined by the concepts of ‘Types of violence’ and ‘Management of coexistence’, respectively (Table 4).

## 3. Procedure

This study was part of the FONDECYT Regular 1191956 project “Family and school education: Emotional socialization in contexts of social and cultural diversity”, reviewed and approved by the Research Ethics Committee of the Catholic University of Temuco (Chile), authorization number 18/19.

Access was negotiated with the centers that finally participated in the study. The parents or guardians of the participants received an e-mail link from their school with information about the study. This included informed consent and a confidentiality notice, explicitly stating that participation was voluntary and that the anonymity and confidentiality of information were guaranteed.

The above mentioned also informed the parents or guardians about the characteristics of the research, the instrument and the time needed to respond. The participating schools gave an average of two weeks for the parents to send their responses, which were collected via a web form. The veracity of the information was guaranteed, given that the parents indicated their e-mail address, identity card number, school, grade and name of their child.

In addition to the information shared with their parents, the students received information about the study during their classes. The instrument’s web form contained all the information referred to in the confidentiality notice given to the parents, as well as an informed consent form, which clarified the voluntary nature of the participation and the possibility of leaving the questionnaire at any time.

The questionnaire was administered during class time, as an activity aimed at discussing violence and school coexistence. The students answered the instrument in web format in the computer room of their school center, together with the respective class, in an average time of 15 min and in the company of a teacher or professional in charge of school coexistence.

The role of the adult in charge was to instruct the students and ensure that the process was carried out in an environment without interruptions. Those in charge of applying the questionnaire ensured that the students who went to the aforementioned room had their parents’ authorization.

All the information was recorded in digital format, maintaining confidentiality and establishing that the records would be eliminated within a period of no more than four years.

Therefore, the procedure was conducted according to the international deontological guidelines referred to in the Declaration of Helsinki and the Singapore Declaration [60,61], as well as those referred to in Chile, by Law 20120 [62].

### Plan for Analysis

The questionnaire was applied at two points in time (pandemic and post-pandemic), with the total sample being divided into two; the first sample was used to explore the abbreviated model and the second was used to validate it.

Considering the characteristics of the instrument, demonstrated both in the development and validation of Muñoz et al. [11], as well as in the re-specification presented by Muñoz et al. [34], in addition to the objective of obtaining an abbreviated model, the inter-judgment of experts was taken as the first element of validity. This is a widely used strategy with many advantages [63], such as determining specific, complex and little-studied constructs [64]

The normality of all the items was evaluated by means of the Kolmogorov–Smirnov test, which allowed us to choose the data analysis methods. The Confirmatory Factor Analysis (CFA) considered using the polychoric correlation matrix and the unweighted least squares mean and variance (ULSMV). The Chi-squared statistic was expected to have a ratio between it and its degrees of freedom of less than 3:1.

Subsequently, goodness-of-fit indicators, such as the root mean square error of approximation (RMSEA), the comparative fit index (CFI) and the Tucker–Lewis index (TLI), were estimated. The RMSEA is considered excellent if the value is less than 0.05 and the CFI is optimal if the values are greater than 0.95 [65,66]. However, the CFI is accepted with values from 0.9 to 0.95 if the RMSEA value is equal to or less than 0.05 [67,68]. For the TLI, values greater than 0.95 are considered ideal and values greater than 0.9 are considered acceptable [67,68,69].

The convergent validity of the scales was evaluated following the proposal of Hair et al. [70], which considers the fulfillment of three conditions for each dimension: (1) standardized loadings with values greater than 0.5 and a statistical significance level with a *p*-value less than 0.05; (2) average variance extracted (AVE), whose values must be greater than 0.5; and (3) composite reliability with values greater than 0.7.

The reliability of the scale was evaluated by means of the McDonald [71] omega coefficient, where values greater than 0.65 are admissible, values greater than 0.7 are acceptable, values between 0.8 and 0.9 are good, and values equal to or greater than 0.9 are excellent [72].

Discriminant validity was reviewed using the criteria of Fornell and Larcker (1981), which consists of comparing the shared and the extracted variance. Thus, if the AVE of a latent variable is greater than the square of the correction between it and the other dimensions, discriminant validity will be demonstrated [70].

In the absence of discriminant validity among the factors and evidence of convergent validity among them, a model with second-order factors is specified [73]. However, following these authors, it should be noted that, since there are high correlations between the first-order factors, a model that merges them can be re-specified, provided that the result of this model is significant. If this model is not significant, the first-order latent variables that show convergent validity will form a second-order factor, without modifying the assumptions made with respect to the first-order CFA measurement model [73,74].

In order to evaluate the structure of the instrument for the gender, ethnicity, dependence and course categories, we performed the following: (1) CFA to each group of categories; and (2) measurement invariance analysis. These analyses were performed following the criteria established by Cheung and Rensvold [75] and Chen [76]. The authors state that configural invariance exists if the goodness-of-fit indicators meet the criteria of the CFA. Metric invariance exists if the variation in the RMSEA is less than 0.015 and/or the variation in the CFI is less than 0.01, with respect to the values of the configural invariance. Meanwhile, scalar invariance is tested if the variance in the RMSEA is less than 0.015 and/or the variance in the CFI is less than 0.01, with respect to the values of the metric invariance. Considering the small number of Mapuche students with respect to the total sample, it was considered that, for sample sizes with unequal groups smaller than 300, the variation in the CFI should be equal to or less than 0.005, and the change in the RMSEA should be equal to or less than 0.010.

The Mann–Whitney U test was used to compare the average scores [77] of the Mapuche and non-Mapuche groups to check for differences in school violence factors.

In order to provide an interpretable scale to the educational institutions, we calculated cut-off points by means of k-means cluster analysis, determining three clusters to differentiate high, medium and low levels in each first-order factor and in the second-order factors.

Finally, measurement invariance analysis and a comparison of the average ranges between the two samples comprising the study were performed.

The analyses were carried out using Microsoft Excel [78], SPSS v.23 [79], RStudio [80], Mplus v.8.1 [81] and G*Power [82].

## 4. Results

The inter-judgment of experts implied the retention of three items per factor and an additional latent variable was proposed for Coexistence Management. Given that the nine-factor model did not show discriminant validity among all the latent variables and that the factor fusion showed inadequate goodness-of-fit indices, a model with second-order factors was chosen (Table 5).

Figure 1 shows the loadings of the first-order factors to their second-order factor.

In samples 1 and 2, the Kolmogorov–Smirnov test (*p* < 0.001) shows that the data do not follow a normal distribution. Sample 1 shows that the proposed abbreviated model has a good fit to the data (Table 6), which allowed us to continue with the analysis of sample 2 and verify that the fit was also adequate.

The following presented results correspond to sample 2. Table 7 shows that all the scale indicators have saturations greater than 0.7, with statistical significance (*p* < 0.001), and that each factor shows a composite reliability ω > 0.8 and an AVE > 0.5, which is evidence of convergent validity. The instrument has good reliability indicators, as each latent variable showed values ω ≥ 0.8.

The goodness-of-fit indices in the CFA for each group of the defined categories showed that the proposed model had a good fit to the data. The measurement invariance analysis, in all categories, showed good indicators and variations in the RMSEA, CFI and TLI that were not relevant, which allowed us to achieve scalar invariance (Table 8).

As shown in Table 9, Mapuche students perceive more physical and verbal violence and social exclusion than non-Mapuche students. The statistically significant differences are in physical violence and social exclusion, with a small effect size.

The cut-off points for the first- and second-order factors are presented in Table 10.

The model that includes the direct effect of Coexistence on Violence has a good fit to the data (RMSEA = 0.044; CFI = 0.946; TLI = 0.940), and the effect of Coexistence on Violence is statistically significant, but of a very small size (γ = 0.08X; *p* < 0.05). Due to the very small effect size, it was decided that the fit of this model would be compared with the model specifying an effect equal to 0, which presents a better fit to the data (RMSEA = 0.035; CFI = 0.967; TLI = 0.963). The delta CFI (ΔCFI = −0.021) indicates that the improvement in the model fit is statistically significant. Therefore, it is established that in the data, the effect of Coexistence on Violence is null.

The goodness-of-fit indices for the unification of both samples and the pandemic and post-pandemic samples were adequate. As shown in Table 11, the model satisfies even scalar invariance between samples 1 and 2.

Table 12 shows the existence of statistically significant differences between the pandemic and post-pandemic samples with a large effect size, where more violence is perceived in the period of on-site classes.

## 5. Discussion and Conclusions

Invasion and colonization is an experience shared by all the world’s indigenous peoples, the effects of which, for those who survived, have endured to the present day [83,84]. In Chile, the Mapuche people have resisted and struggled with different nuances at different times [30], without mitigating the consequences of two hundred years of usurpation, discrimination and domination perpetrated by the Chilean state [34].

Therefore, it is important for research on violence and coexistence management to consider the cultural variations in emotional development in spaces of sociocultural diversity. In this regard, the present study provides a scale that can be applied in educational spaces with a high percentage of Mapuche students.

Statistically significant differences were found in the perception of verbal violence and social exclusion between Mapuche and non-Mapuche students, where it is the indigenous subject who perceives more violence. Likewise, Mapuche students perceive more verbal violence, although no statistical significance was found; these results correlate with the study of Muñoz-Troncoso et al. [34]. The findings of the present research could be related to the belief system revealed in the research of Riquelme et al. [36] on emotional development in the Mapuche culture, where the school system does not consider the cultural variations between Mapuche and non-Mapuche students. This would originate from the models of upbringing the students are exposed to, such as learning to listen and observe attentively, by which students are categorized as introverted or shy, falling into assumptions or prejudices [44].

Such variations in the perception of violence in indigenous students is related to the principles of Mapuche family education, where respect, kindness and generosity are fundamental components in the formation of new generations [38,39,43]. It is feasible to assert that the history of suffered violence, discrimination and marginalization has led to the construction of different parameters in the understanding of violence, its dynamics and expression in educational scenarios between Mapuche and non-Mapuche students. Thus, Mapuche students perceive the dynamics of racist violence rooted in the social and school structure, a product of the colonial roots of the school and the national education system; this constitutes a basis for building their personal and socio-cultural identity in opposition to the hegemonic society [85].

The ancestry category represents a group that indicates not knowing whether they are Mapuche or not; this has also been reported in previous applications of the full version of the instrument [10,11,34]. While it is impossible to determine why they do not know whether they belong to the Mapuche people or not, it is striking that official reports [31] about the percentage of Mapuche children within this age range in the city of Temuco are higher than those found in the analyzed samples. There are not enough arguments available to determine whether they are really Mapuche students who do not self-identify as such; however, this assumption is somewhat supported by the concept of ethnic shame. This refers to the construction of a collective imaginary in which children, youth and adults have decided to ignore their ethnic–cultural origin in order to assume themselves as part of the hegemonic society. This, historically, would have implied the abandonment of their own educational model, inducing the loss of indigenous surnames and the construction of a family narrative in denial of the indigenous component and that overvalues the European root [43,50,86].

In relation to the reports regarding complaints in the area of school coexistence referred to by the Mineduc [87], the present study shows statistically significant differences between pandemic and post-pandemic students, the latter being those who perceive greater violence in all the factors of the scale. This allows us to sustain that the dynamics of coexistence in the school give rise to conflicts, which are diminished in periods of confinement [4,7].

The cut-off scores were calculated based on the participants of the post-pandemic sample; however, they can be used to interpret the results obtained from the application of the instrument in municipal or subsidized private schools in the commune of Temuco.

Regarding the effect that the perception of Coexistence Management can have on the perception of School Violence, the analysis using Structural Equation Models was null. The effect found in the present study could be explained by the fact that the proposed measurement model is relatively simple, and that there could be mediating and/or moderating variables, including other exogenous variables that could explain the variability in violence.

In response to the proposed objectives, the present study provides a valid and reliable measurement instrument, which makes possible an unbiased comparison between the groups of the defined categories. At the same time, it makes available a scale contextualized to the Chilean reality, particularly to the region of La Araucanía, whose application is relevant in schools with similar characteristics to those that participated in the present study. The application of this instrument and its results in Chilean schools represents a relevant input to be integrated into school coexistence management plans. These actions, besides being important to evaluate, prevent, reduce violence and improve the management of school coexistence, are part of the actions required by Chilean public policy.

The results allow us to review the hypotheses put forward:

**H1.** 
*The abbreviated version of the scales has good goodness-of-fit indicators and indicators of reliability and validity.*


This hypothesis is confirmed, since the fit of the proposed model to the data, reliability and validity are tested.

**H2.** 
*The proposed scale has the same meaning for the gender, ethnicity and dependence categories.*


This hypothesis is confirmed, since scalar invariance was evidenced in the defined categories.

**H3.** 
*There are statistically significant differences in the perception of violence between Mapuche and non-Mapuche students.*


This hypothesis is accepted, since the statistical significance is less than 0.05 in the factors of verbal violence and social exclusion.

**H4.** 
*There are statistically significant differences between pandemic (online, face-to-face and hybrid classes) and post-pandemic (face-to-face classes) participants.*


This hypothesis is confirmed, since the statistical significance was evidenced at the 0.01 level.

**H5.** 
*There is no causal relationship of the results of Coexistence Management on School Violence.*


This hypothesis is confirmed, since a null effect was evidenced.

In view of the above, this study may enable us to observe the effects of the implementation of the colonial project in a Mapuche context, where the historical violence experienced for centuries affects both the perceptions of violence developed in the school environment and the social dynamics of interaction between people from different socio-cultural origins. The group of students who state that they do not know their ethno-cultural origin are particularly striking, since this could be evidence of the effects of the uprooting induced by the colonialist school project developed by the Chilean State in Mapuche territory. A survey conducted by the Center for Public Studies (CEP) among Mapuche subjects showed that 11% feel Chilean, 17% consider themselves Mapuche, 45% define themselves as Chilean and Mapuche at the same time, 21% consider themselves Mapuche first and Chilean later, and 6% consider themselves Chilean first and Mapuche later [88]. In addition, the results showed statistically significant differences with respect to the 2006, 2016 and 2022 surveys, with an increase in the group that identified itself as Chilean and Mapuche at the same time, rising from 28% to 39% and 45%.

It can be argued that addressing the dynamics of social violence reproduced in the school environment is feasible via the implementation of a critical intercultural education model that favors the relativization of power relations and the status of societies, groups and people in contact with each other [89]. This project of critical interculturality requires schools to generate actions to contribute at the social level in the fight against racism, colonialism, and their expressions and roots, contributing to generating structural and sustained social changes over time. This will have a potential impact on the mental health of students involved in school violence, as improve learning outcomes.

The study can be furthered by broadening the age range of students, or by replicating it specifically in high-school students. Future studies may include a comparison at the present time, with face-to-face classes being fully reinstated in the school system. Also, the replication of the study may consider a design that aims to understand the characteristics of the group that reports not knowing whether or not they belong to the Mapuche people.

The main limitation of this study is related to the size of the sample, which, not being representative of the Chilean population, cannot generalize the results to the country’s school system. Also, the data do not include information on the students’ mental health and/or learning outcomes, which would allow us to investigate these factors’ correlation with the perception of school violence and coexistence management.

Finally, it is possible to affirm that we employed an abbreviated instrument that is appropriate for measuring types of school violence and coexistence management using students’ perceptions.

## Figures and Tables

**Figure 1 behavsci-13-00686-f001:**
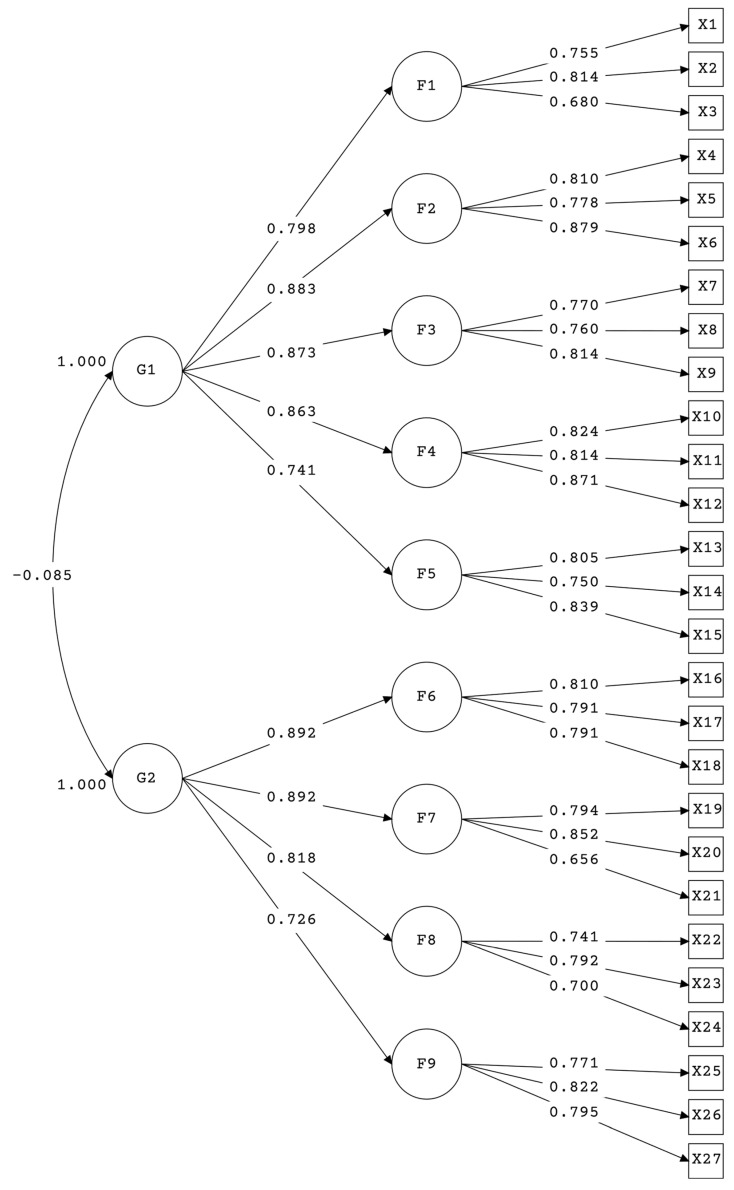
Path model of the scale. Source: prepared by the authors.

**Table 1 behavsci-13-00686-t001:** Characterization of participants. Source: prepared by the authors.

Categories	Groups	Pandemic *		Post-Pandemic	
		Frequency	Percentage	Frequency	Percentage
Gender	Men	420	49.5	540	52.8
	Women	428	50.5	482	47.2
Nationality	Chilean	840	99.1	1002	98.0
	Foreign	8	50.5	20	2.0
Ethnicity	Does not belong	494	58.3	542	53.0
	Does not know	106	12.5	348	33.5
	Mapuche	248	29.2	138	13.5
Dependency	Municipal	372	43.9	576	56.4
	Subsidized private	476	56.1	446	43.6
Schools	Municipal 1	190	22.4	282	27.6
	Municipal 2	182	21.5	294	28.8
	Part. Subv. 1	211	24.9	209	20.5
	Part. Subv. 2	265	31.3	237	23.2
Course	5th grade	188	22.2	269	26.3
	6th grade	227	26.8	243	23.8
	7th grade	187	22.1	275	23.9
	8th grade	246	29.0	235	23.0
Classroom mode	Online	662	78.1	0	0.0
	On-site	46	5.4	1022	100.0
	Online and on-site	140	16.5	0	0.0

* Note: The pandemic sample corresponds to the period of ‘gradual return’ to face-to-face attendance.

**Table 2 behavsci-13-00686-t002:** Structure of the CENVI questionnaire. Source: Muñoz-Troncoso et al. [34].

Factors of Second Order	First-Order Factors (Dimensions)		Abbreviation	Items Per Dimension	Item No.
Factor 1 Types of Violence	1	Verbal violence	VV	8	1–8
	2	Physical violence	PV	11	9–19
	3	Social exclusion	SE	10	20–29
	4	Digital violence	DV	9	30–38
	5	Teacher violence	TV	8	39–46
Factor 2 Coexistence management	1	Education	ED	10	47–56
	2	Assurance	AS	10	57–66
	3	Participation	PA	8	67–74

**Table 3 behavsci-13-00686-t003:** Convergent validity and reliability indicators. Source: Muñoz-Troncoso et al. [34].

Factors	Dimensions	Range Correlations Minimum; Maximum	Reliability Composed	AVE
Coexistence	Verbal violence	0.578; 0.829	0.894	0.516
	Physical violence	0.615; 0.873	0.926	0.536
	Social exclusion	0.595; 0.831	0.912	0.511
	Digital violence	0.728; 0.848	0.943	0.650
	Teacher violence	0.570; 0.838	0.895	0.521
Violence	Education	0.582; 0.804	0.912	0.510
	Assurance	0.643; 0.805	0.917	0.528
	Participation	0.623; 0.782	0.892	0.510

Note. All items had a statistical significance of *p* < 0.001.

**Table 4 behavsci-13-00686-t004:** Actions related to the scale factors. Source: Muñoz-Troncoso et al. [34].

Factors of Second Order	Factors of First Order	Abbreviation	Definition
School Violence	Verbal Violence	VV	Aggression by means of words, such as insults, threats, offensive nicknames.
	Physical Violence	PV	Pushing and shoving, hair pulling, pinching, punching, kicking or hitting with objects. It is indirect physical violence when it is perpetrated on the victim’s belongings or work materials.
	Social Exclusion	SE	Acts of discrimination or rejection based on academic performance, nationality, cultural or ethnic differences, physical characteristics or personal appearance.
	Digital Violence	DV	Aggression via cell phones or other internet communication devices, by means of photos, videos or text messages.
	Teacher Violence	TV	Aggression by the teacher towards the student, either verbal, physical or discriminatory.
Coexistence Management	Education	ED	Practices for reflection and education based on dialogue, respect and legitimate acceptance of the other, in order to reduce the risk of violent situations.
	Assurance	AS	Construction and enforcement of coexistence rules to prevent, control and sanction violence.
	Participation	PA	Actions aimed at the integration of the members of the educational community, to contribute to the construction of safe spaces free of abuse.

**Table 5 behavsci-13-00686-t005:** Structure of the abbreviated School Violence and Coexistence Management Scale. Source: prepared by the authors.

	Factors of Second Order		Factors of the First Order	Abbreviation	Item No.
G1	School Violence	F1	Verbal violence	VV	X1–X3
		F2	Physical violence	PV	X4–X6
		F3	Social exclusion	SE	X7–X9
		F4	Digital violence	DV	X10–X12
		F5	Teacher violence	TV	X13–X15
G2	Coexistence Management	F6	Reflection	RE	X16–X18
		F7	Education	ED	X19–X21
		F8	Assurance	AS	X22–X24
		F9	Participation	PA	X25–X27

**Table 6 behavsci-13-00686-t006:** Fit indicators of the abbreviated model in both samples. Source: prepared by the authors.

Sample	N	X2	DF	*p*-Value	RMSEA	IFC	TLI
Sample 1	848	750.451	314	<0.001	0.041	0.955	0.950
Sample 2	1022	944.023	314	<0.001	0.044	0.946	0.940

**Table 7 behavsci-13-00686-t007:** Convergent validity and reliability indicators. Source: prepared by the authors.

Factors of Second Order	Factors of First Order	Range Correlation Minimum–Maximum	Reliability Composed	AVE
School Violence	VV	0.680–0.814	0.8	0.565
	PV	0.778–0.879	0.9	0.678
	SE	0.760–0.814	0.8	0.611
	DV	0.814–0.871	0.9	0.700
	TV	0.750–0.839	0.8	0.638
Coexistence Management	RE	0.791–0.810	0.8	0.636
	ED	0.656–0.852	0.8	0.596
	AS	0.700–0.792	0.8	0.555
	PA	0.771–0.822	0.8	0.634

Note. All items had a statistical significance of *p* < 0.001.

**Table 8 behavsci-13-00686-t008:** Contrast in measurement invariance of categories. Source: prepared by the authors.

Category	Level	RMSEA	IFC	TLI	ΔRMSEA	ΔCFI	ΔTLI
Gender	Configural	0.044	0.946	0.940	-	-	-
	Metrics	0.043	0.947	0.943	0.001	0.001	0.003
	Scalar	0.042	0.945	0.945	0.001	0.002	0.002
Ethnicity	Configural	0.045	0.944	0.937	-	-	-
	Metrics	0.043	0.944	0.941	0.002	0.000	0.004
	Scalar	0.039	0.950	0.951	0.004	0.006	0.010
Dependency	Configural	0.042	0.947	0.940	-	-	-
	Metrics	0.041	0.947	0.943	0.001	0.000	0.003
	Scalar	0.042	0.942	0.942	0.001	0.005	0.001
Course	Configural	0.044	0.944	0.937	-	-	-
	Metrics	0.043	0.943	0.940	0.001	0.001	0.003
	Scalar	0.043	0.937	0.940	0.000	0.006	0.000

**Table 9 behavsci-13-00686-t009:** Differences in the perception of violence between Mapuche and Non-Mapuche students. Mann–Whitney U test. Source: prepared by the authors.

	Ancestry						
	Non-Mapuche	Mapuche					
	*n* = 542	*n* = 138					
Variables	Average	Average	Z	U	*p*	1 − β	*d*
	Range	Range					
School Violence							
VV	332.98	370.03	−1.992	33,323.5	0.046	0.52	0.20
PV	338.52	348.28	−0.527	36,324.0	0.598	0.71	0.08
SE	329.80	382.51	−2.838	31,600.0	0.005	0.52	0.28
DV	341.70	335.79	−0.324	36,848.0	0.746	0.75	0.02
TV	345.28	321.71	−1.294	44,396.0	0.196	0.38	0.09

**Table 10 behavsci-13-00686-t010:** Levels according to the first and second-order factors. Source: prepared by the authors.

Factor	Under	Medium	High
VV	3–6	7–9	10–12
PV	3–4	5–8	9–12
SE	3–5	6–9	10–12
DV	3–5	6–9	10–12
TV	3–5	6–8	9–12
RE	3–5	6–9	10–12
ED	3–5	6–8	9–12
AS	3–5	6–9	10–12
PA	3–5	6–9	10–12
Violence	15–28	29–40	41–60
Coexistence	12–25	26–35	36–48

**Table 11 behavsci-13-00686-t011:** Measurement invariance contrast in pandemic and post-pandemic sample. Source: prepared by the authors.

	LEVEL	RMSEA	IFC	TLI	ΔRMSEA	ΔCFI	ΔTLI
Sample 1 and 2	Configural	0.042	0.951	0.945	-	-	-
	Metrics	0.041	0.950	0.946	0.001	0.001	0.001
	Scalar	0.042	0.945	0.945	0.001	0.005	0.001

**Table 12 behavsci-13-00686-t012:** Differences in the perception of violence between pandemic and post-pandemic. Mann–Whitney U test. Source: prepared by the authors.

	Period						
	Pandemic	Post-Pandemic					
	*n* = 848	*n* = 1022					
Variables	Average	Average	Z	U	*p*	1 − β	*d*
	Range	Range					
School Violence							
VV	733.66	1102.97	−14.519	262,169.5	<0.001	1	0.72
PV	721.30	1113.23	−15.911	251,684.5	<0.001	1	0.76
SE	761.68	1079.72	−12.835	285,932.0	<0.001	1	0.58
DV	734.30	1102.45	−15.670	262,707.0	<0.001	1	0.72
TV	749.60	1090.20	−14.760	275,224.0	<0.001	1	0.67

## Data Availability

Data available at [90] and at the Appendix A.

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
