# Peer review of "Validation of an Abbreviated Scale of the CENVI Questionnaire to Evaluate the Perception of School Violence and Coexistence Management of Chilean Students: Differences between Pandemic and Post-Pandemic"

_behavsci, 2023, doi:10.3390/bs13080686_

Round 1
Reviewer 1 Report
The goal of this study was to specify an abbreviated model of the CENVI questionnaire for students from 5th to 8th grade (10 to 13 years old), in order to know the perception of violence and management of school coexistence, with emphasis on the differences between Mapuche and non- Mapuche students. This topic is very important and relevant, because it school context (history of invasion and colonization) is relevant factor as for violence, and especially management of school coexistence. Furthermore, congratulations authors for selection important topic, minority (and majority) experience in school, and recognition of the importance of looking at problems from different perspectives (Mapuche and non- Mapuche students). But still there are some shortcuts of study. Generally, the manuscript suffers from a lack of focus on the research goal, and have a lot of content that is less important for this topic. So, the appeal to the authors is to direct the text a little more towards the goal of the research.
First of all, the abstract should be a total of about 200 words maximum, and this abstract is over that number. And, because goal of manuscript is to specify abbreviate model od CENVI questionnaire, it seems that too much space in abstract is provide to description of covid and non-covid context. Furthermore, it is unclear what CENVI stands for? Abbreviations should be defined the first time they appear.
Although context of school system is very important in terms of school violence, it is recommendation to short of Introduction part, and statistical data from that part and to emphasize description school context regarding differences between Mapuche and non- Mapuche students. So, in passus School spaces with social and cultural diversity is needed more detailed description of Chilean school system, and distinction from Mapuche and non-Mapuche students. Furthermore, in the same passus same content is repeated few times, for example:
Muñoz-Troncoso Research [46] revealed the existence of statistically significant differences in the perception that Mapuche and non-Mapuche students had of physical and verbal violence, and that the indigenous student is the one who perceived higher levels of violence.
“A recent study by Muñoz-Troncoso et al. [46] explored the differences in perceptions of violence between Mapuche and non-Mapuche children, showing statistically significant differences, revealing that Mapuche students perceive more physical and verbal violence than non-Mapuche children.].
Therefore, author(s) are asked to carefully read the text and to exclude redundant parts.
Furthermore the sentence “The scale will provide a measurement model with the potential mobility from the 261 scientific to the empirical.“ is unclear.
It is, also, unclear why authors include Coexistence Management and School Violence in same scale if they hypnotize that: „H5: There is no causal relationship of the results of Coexistence Management on School 282 Violence.“
Furthermore, in section Participants, I have question-“Are participants from age 9-14 children and young people, or they are “children and adolescents””? Which categorization are used by author(s)?
„Sample 1 (Pandemic, accessed in December 2021) is composed of 848 children and
288 young people from the commune of Temuco (Chile), aged 9 to 14 years (M=11.90; SD=1.27)
289 who attended from 5th to 8th grade in 4 schools, 2 subsidized private and 2 municipal. „
In table 2., instead term Genre use term gender!
Numbers in Table 2 are not logical -about Nationality!
Also, in Table 2, what is difference between 5th Grade and Basic 6th grade? Why ih 6th grade bassic, and 5th, 7th and 8ht are not basic?
In Instruments, is commonly to first described instruments, and after to write some additional information for example „In both applications, the full version of the school coexistence questionnaire for non299 violence (CENVI) was used [46] which explores students' perceptions regarding types of 300 school violence and coexistence management (Table 3). The above, in the logic of a divi301 sion of the total sample, where the first sample was used to explore the abbreviated model 302 and the second to validate it. The study by Muñoz-Troncoso et al. [46] showed evidence in favor of different ele307 ments of validity and reliability. The instrument has good fits of the proposed model to 308 the data: X2=7304.699; DF=2618; RMSEA=0.036; CFI=0.914; TLI=0.911. Factor 1 dimensions 309 have correlation measures from 0.6 to 0.9 and factor 2 dimensions have correlations from 310 0.7 to 0.8. All items have factor loadings greater than 0.5 with statistical significance 311 (p<0.001). The composite reliability for each dimension is ω >0.89. The average variance 312 extracted (AVE) shows values greater than 0.5. Finally, the reliability indicators are opti313 mal, since the dimensions have a value ω ≥0.9. Some details of the above are shown in 314 Table 4.“ So, it is necessary to reorganize that paragraph.
The sentence “The above, in the logic of a divi301 sion of the total sample, where the first sample was used to explore the abbreviated model 302 and the second to validate it.“ should be moved to Procedure.
I am not sure, are data in Table 4 correct!
Where authors have preregistered studies or analysis plans, links to the preregistration must be provided in the manuscript.
In Discussions and Conclusions a lot of space is dedicated to context of colonization, and comment to results of this research is missing. Also it is unclear sentence „It was stated that there is evidence of little or no incidence of students and parents.“-it is unclear when the parents were questioned.
In this form of manuscript (there is too many (redundant) tables, the text is not in focus-reorganization is needed, Instrument is unclear, Plan for Analysis it seems to extensive) publishing of this text is not recommended.
Reviewer 2 Report
Overall, the paper addresses an important and timely topic regarding the resumption of face-to-face classes in Chile's school system after an extended period of online activities and hybrid learning. The study aims to investigate the perception of violence and management of school coexistence among Mapuche and non-Mapuche students, specifically focusing on the differences between the two groups. While the study provides valuable insights, there are a few areas that could benefit from further clarification and improvement.
The introduction is relatively long. It could be shortened by better synthesizing the literature. It also needs to clearly state the research objectives and the significance of investigating the differences between Mapuche and non-Mapuche students in relation to school violence and coexistence management.
Also an abbreviated model of the CENVI questionnaire was used, but it is not clear what are the limitations of the original questionnaire, its validity, and reliability. A brief description of the questionnaire and its development would enhance the reader's understanding of the measurement tool.
In methods, the sample includes children. Please delete the words ‘young people’.
The data collection procedure is not clear. I understand that parental consent was obtained but how? How many schools were targeted? Where they have been located?. Also, how were the classes within schools selected? Then how were the children selected? Any inclusion or exclusion criteria? When the child returned the signed consent, how did s/he contact the researchers to return them? How many days did students and their parents have to fill in the consents and questionnaires? How did authors ensure that children did not discuss their responses with their peers, and others? Were teachers of researchers setting in class while children were filling responses?
Actually, the fact that data collection was conducted in class may impose some coercion. How did researchers protect children’s rights not to participate?
The students answered qs on a web format. But how? Each student was handled a device? Or did they use their phones? How was this process controlled?
Did researchers obtain children assents? How?
Both the results and discussion sections over emphasize the statistical significance without consideration of clinical significance of the findings. Please include the effect sizes for all inferential statistics.
Limitations: While the conclusion briefly mentions the limitation of the sample size and lack of representativeness, it could expand on the potential implications of these limitations.
none
Round 2
Reviewer 1 Report
After Authors reorganized the manuscript (Validation of an abbreviated scale of the CENVI questionnaire to evaluate the perception of school violence and coexistence management of Chilean students: Differences between pandemic and post-pandemic), it has contributed to greater clarity, and in this version of manuscript is suitable for publication.
Reviewer 2 Report
I am pleased to acknowledge that all of my initial concerns have been appropriately addressed in this revised manuscript. The revisions have effectively enhanced the overall quality of the paper, making it a more comprehensive and impactful contribution to the field.
The concise and refined introduction now clearly states the research objectives and the significance of investigating the differences between Mapuche and non-Mapuche students in relation to school violence and coexistence management. The introduction effectively synthesizes the literature, making it more engaging for the reader.
The expanded description of the data collection procedure, including information about parental consent, school selection, class and student selection, and the control mechanisms for maintaining confidentiality, ensures a more transparent methodology.
Once again, I would like to thank the authors for their dedication to improving the work.